# Assessment of Red Sea temperatures in CMIP5 models for present and future climate

**Miguel Agulles[1]\*, Gabriel Jordà[1], Ibrahim Hoteit[2], Susana Agustí[2], Carlos M. Duarte[2]**

**1** Centre Oceanogràfic de Balears, Instituto Español de Oceanografía, Palma, Spain, **2** King Abdullah University of Science and Technology (KAUST), Red Sea Research Center (RSRC), Thuwal, Saudi Arabia

\* miguel.agulles@ieo.es

**Data Availability Statement:** All relevant data are within the manuscript and its Supporting Information files.

**Funding:** This research was funded by King Abdullah University of Science and Technology

## Abstract

The increase of the temperature in the Red Sea basin due to global warming could have a large negative effect on its marine ecosystem. Consequently, there is a growing interest, from the scientific community and public organizations, in obtaining reliable projections of the Red Sea temperatures throughout the 21$^{st}$ century. However, the main tool used to do climate projections, the global climate models (GCM), may not be well suited for that relatively small region. In this work we assess the skills of the CMIP5 ensemble of GCMs in reproducing different aspects of the Red Sea 3D temperature variability. The results suggest that some of the GCMs are able to reproduce the present variability at large spatial scales with accuracy comparable to medium and high-resolution hindcasts. In general, the skills of the GCMs are better inside the Red Sea than outside, in the Gulf of Aden. Based on their performance, 8 of the original ensemble of 43 GCMs have been selected to project the temperature evolution of the basin. Bearing in mind the GCM limitations, this can be an useful benchmark once the high resolution projections are available. Those models project an averaged warming at the end of the century (2080–2100) of 3.3 ±> 0.6˚C and 1.6 ±> 0.4˚C at the surface under the scenarios RCP8.5 and RCP4.5, respectively. In the deeper layers the warming is projected to be smaller, reaching 2.2 ±> 0.5˚C and 1.5 ±> 0.3˚C at 300 m. The projected warming will largely overcome the natural multidecadal variability, which could induce temporary and moderate decrease of the temperatures but not enough to fully counteract it. We have also estimated how the rise of the mean temperature could modify the characteristics of the marine heatwaves in the region. The results show that the average length of the heatwaves would increase ~15 times and the intensity of the heatwaves ~4 times with respect to the present conditions under the scenario RCP8.5 (10 time and 3.6 times, respectively, under scenario RCP4.5).

## Introduction

The Red Sea holds one of the most diverse marine ecosystems in the world, but fragile and vulnerable to oceanic warming [1], as reflected in massive coral bleaching across the southern Red Sea during the 2015 global bleaching event [2–4]. Water temperature plays a key role in

(KAUST) through funds provided to S.A. and C.M.D (BAS/1/1072-01-01). M.A. has been partly funded by the European Union's Horizon 2020 research and innovation programme under grant agreement No 776661 (SOCLIMPACT project), and "Proyecto Intramurales Especial REDTEM - CSIC (202030E234)" Each named author has substantially contributed to conducting the underlying research and drafting this manuscript. Additionally, to the best of our knowledge, the named authors have no conflict of interest, financial or otherwise.

**Competing interests:** The authors have declared that no competing interests exist.

the performance of marine organisms and the regulation of ecosystems processes, which are usually acclimated to the local thermal range. Hence, changes in the temperature field (e.g. warming) affect marine organisms, typically by shifting their distribution poleward and advancing their phenology [5,6]. Besides the long-term changes, extreme events like marine heatwaves (MHWs), defined as prolonged periods of anomalously high temperatures, can have severe impacts on marine ecosystems. For instance, coral bleaching in tropical ecosystems [2], reduced surface chlorophyll levels due to increased surface layer stratification, mass mortality of marine invertebrates due to heat stress, and fishery collapse [7,8]. Semi-enclosed seas are particularly vulnerable to ocean warming, which is often amplified in these seas [9], and where the scope for organisms to adapt by migrating poleward is restricted by the presence of land masses [10]. This is the case of the Red Sea, which is considered to be almost closed in its northern boundary in terms of water exchanges, although the Suez Canal has offered a limited escape route into the Mediterranean, where many Red Sea species are becoming established [11]. Consequently, there is an urgency to develop reliable thermal projections throughout the 21$^{st}$ century for marine ecosystems, particularly so for semi-enclosed seas. Although being one of the global hot spots for climate-vulnerable coral reefs, the assessment of these projections is limited to few works. One of them use CMIP3 GCMs without doing any model assessment or selection before providing future temperatures [12]. The other one projects the future temperature in the Red Sea using an unique GCM after a rather simple selection procedure [13].

Long-term temperature projections are usually based on global climate models (GCMs) run under different scenarios of greenhouse gas (GHG) emissions. The most used ensemble of GCMs is the one produced by the fifth phase of the Climate Model Intercomparison Project (CMIP5, [14]), which has been used for a large number of studies of climate change of both the atmosphere and ocean. However, these global models have weaknesses when applied to regional seas [15]. Their coarse spatial resolution (typically from 0.2° to 2° in the ocean) precludes them from solving key processes that may have a leading role in the local response to global warming [16]. But for some processes characterized by simpler dynamics, GCMs may be enough to reproduce the local response to global warming. In the case of the Red Sea, a recent study by [17] has shown that interannual variations of the heat content in the 0–100 m layer are mostly driven by changes in the atmospheric temperature. Thus, GCMs (at least some of them), may be able to represent the heat uptake variability in the region. Therefore, whether GCMs are well suited for a particular region or process must be analysed case by case [18]. In any case it is clear that regional high-resolution simulations partially overcome the shortcomings of global models and are, therefore, expected to be more reliable for regional seas (e.g MedCORDEX in the Mediterranean Sea; [19]). Unfortunately, regional projections for the Red Sea are not available at present, so the only source of projections for the evolution of sea temperature throughout the 21$^{st}$ century is the global simulations.

Here we focus on conducting a skill assessment of the CMIP5 models in the characterization of the Red Sea temperature. Namely, in a first step, different comparisons with an observational dataset are performed aiming at assessing the quality of the models in simulating different aspects of the ocean heat uptake. Additionally, we compare the CMIP5 simulations with two state-of-the-art simulations, a medium resolution global reanalysis [20] and a high resolution regional model [21]. That comparison provides a benchmark for the expected quality of model results in the Red Sea. These analyses allow to discard models showing low skills in reproducing the ocean heat uptake in the region. The models that are kept are then used to analyse the projected 3D temperature evolution under different emissions scenarios. It is clear that GCMs are not the best option for that purpose, but until an ensemble of high resolution ocean models is available, the projections from the selected GCMs can provide a first approximation to it. Lastly, we focus on marine heatwaves (MHWs). The reason is they have a strong

influence on marine ecosystem structure and function [22–24]. They lead to discrete climatic events that can drive step-wise changes in species distributions and ecosystem structure and functioning [25]. Thus, MHWs have profound ecological impacts that included widespread mortality of benthic invertebrates [26] and loss of seagrass meadows [27]. The future evolution of MHWs depends on changes in the mean temperature and on changes in the intra-seasonal variability. The first factor is usually the dominant one, accounting for a large part of the projected changes in MHWs [18,22,23]. Unfortunately, high frequency outputs from the selected GCMs were not available, so here we analyse how changes in the mean temperature would affect the characteristics of MHWs in the Red Sea.

The paper is structured as follows. The observational datasets, the numerical simulations and the indicators used to assess model skills are presented in Section 2. The model skill assessment and model selection are presented in Section 3 while the results for temperature projections are described in Section 4. Then the results are discussed in Section 5 and the main conclusions are drawn in Section 6.

## Data and methods

### Observational datasets

**In-situ data.** We used the TEMPERSEA gridded product [17] as the reference observational dataset for model performance assessment. TEMPERSEA is based on a large dataset of *in-situ* temperature observations which are homogenized and mapped into a 3D field using optimal interpolation. The database covers the Red Sea and the Outer Region with a spatial resolution of 0.25˚x 0.25˚ and 23 vertical levels, (Fig 1 for the limits of both regions), providing

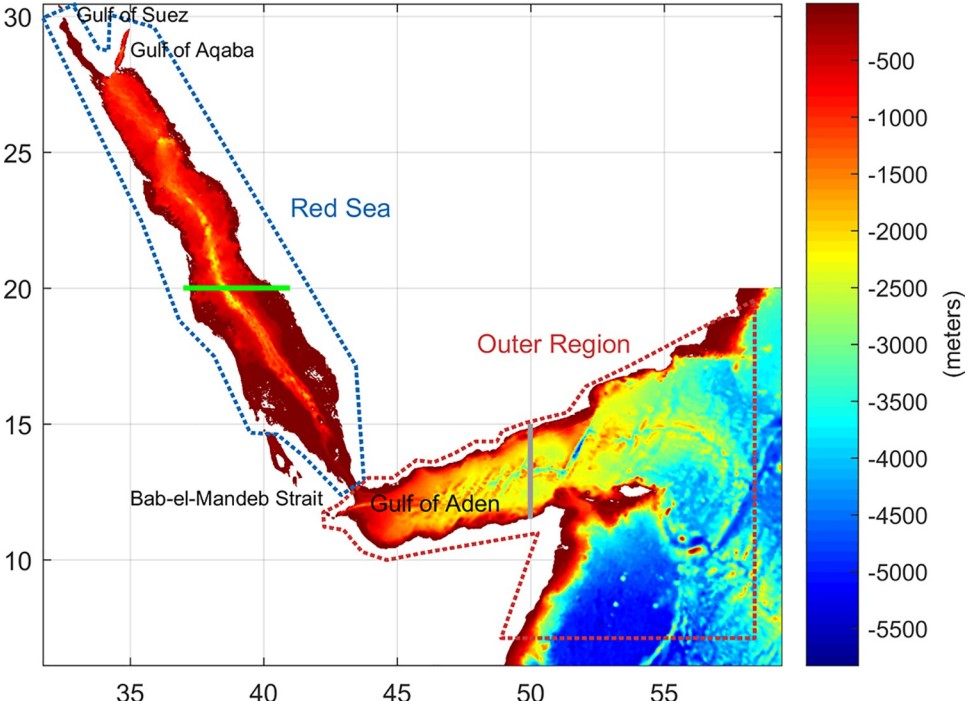

**Fig 1. Domain and bathymetry of the region of interest.** The two zones used in the presentation of the results (Red Sea and the Outer Region) are identified by blue and red dashed contours, respectively. Eastern boundary limit of the KAUST model (vertical grey line). Northern and Southern zones, as considered for the computations, are separated at latitude 20˚ (horizontal green line).

monthly 3D temperature fields for the period 1958–2017. More details on the TEMPERSEA product can be found in https://doi.pangaea.de/10.1594/PANGAEA.909472.

**Satellite data.** The monthly time resolution of TEMPERSEA makes it unsuitable for the characterization of extreme events such as heatwaves. For that purpose, we used daily satellite SST observations which have already been used in the Red Sea [28]. The remote dataset was obtained from the National Ocean and Atmosphere Agency (NOAA), based on AVHRR (Advanced Very High-Resolution Radiometer) imagery over the period 1981–2017. These data have a spatial resolution of 0.25˚x0.25˚ and can be obtained at daily temporal resolution from the National Center for Environmental Information (NCDC-NOAA, https://www.ncdc.noaa.gov/isd/data-access). These data are used to characterize the high-frequency temperature variability in the Red Sea (section 4.3). It must be noted that the agreement between the first layer of the TEMPERSEA product and the satellite SST is very high at monthly time scales [17]. Thus, for clarity, the comparison of GCMs with monthly satellite SST fields has not been included as the results are very similar to those obtained with TEMPERSEA.

**Atmospheric reanalysis.** JRA55 atmospheric reanalysis is used to assess the role of the atmosphere in the sea temperature variations. It has a global coverage with a spatial resolution of 1.25˚ and covers the period 1958–2014, coinciding with the establishment of the global radiosonde observing system [29]. The 3D air temperature fields have been freely downloaded from https://rda.ucar.edu/datasets/ds628.1/?hash=access#!access.

## Numerical simulations

**The CMIP5 ensemble of global climate models.** The ensemble of GCMs from the fifth phase of the Coupled Model Intercomparison Project (CMIP5) was used in this work. The goal of CMIP5 is to produce a set of coordinated climate model experiments that focus on major gaps in understanding past and future changes in climate [14]. The GCMs in the CMIP5 ensemble have a different degree of complexity but all of them include at least an ocean and an atmosphere component interactively coupled. The spatial resolution of CMIP5 models ranges from 0.2˚ to 2˚ in the ocean. From the 49 models initially considered, 6 of them were automatically discarded as their land-sea mask did not include the Red Sea (see S1 Table).

Three different sets of simulations were analyzed. First, the historical simulations that cover the industrial period (mid-nineteenth century to 2005) only forced by observed GHG and aerosol concentrations. As no data assimilation is involved, one cannot expect that the chronology of events matches the observed ones, so we analysed those simulations in terms of statistical moments (i.e., mean and standard deviation). The other two sets of simulations correspond to future projections in which the models are forced with specified atmospheric greenhouse gas concentrations, known as RCPs (Representative Concentration Pathways [14] throughout the 21st century (2005–2100). In particular, we analysed a high emissions scenario (RCP8.5), also known as business-as-usual scenario, and a midrange emissions scenario (RCP4.5), which represents a scenario with moderate mitigation policies [30]. The data were obtained from (https://pcmdi.llnl.gov/mips/cmip5/). In particular, we analyzed ensembles of 43 historical simulations, 26 RCP8.5 simulations and 39 RCP4.5 simulations (see S1 Table).

**Reference state-of-the-art simulations.** In order to have a benchmark of the state-of-the-art quality in Red Sea modelling, two high quality simulations of present climate are used. One is a regional high resolution hindcast (KAUST) and the other is a global reanalysis with medium resolution (GLORYS). The King Abdullah University of Science and Technology (KAUST) Red Sea hindcast is based on the MIT ocean general circulation model [31] tuned to the regional processes of the Red Sea as in [21]. The model domain covers the entire Red Sea

(including the Gulf of Suez and the Gulf of Aqaba) and part of the Gulf of Aden (area enclosed by 10–30˚N and 30–50˚E) but does not cover the Outer Region. The model is forced by ECMWF (European Centre for Medium-Range Weather Forecasts) atmospheric fields [32]. It provides monthly mean temperatures with a spatial resolution of 0.1˚ and 50 vertical z-levels for the period (2001–2017). This model is developed as an updated version of the one implemented by [21] to investigate the general and overturning circulation in the Red Sea. The model outputs were validated against different observational datasets including temperature and have been used in various Red Sea studies. To name but a few, it has been used to analyse the eddy field [33], the phytoplankton phenology in the South of the Red Sea [34] and in the North [35] or the general connectivity [36].

In order to complement the KAUST model and to also cover the Gulf of Aden, the GLORYS.S2V4 global reanalysis was also analyzed [17,20]. This product was generated by NEMOv3.1 ocean model with a horizontal resolution of 0.25˚ and 75 vertical z-levels. It is forced by ERA-Interim atmospheric fields [32], for the period 1993 to 2015. GLORYS assimilates along-track satellite observations of sea level anomaly, sea ice concentration and SST, and *in situ* profiles of temperature and salinity from the CORA data base [37]. The data is available at (http://marine.copernicus.eu/services-portfolio/access-to-products/?option=com_csw&view=details&product_id=GLOBAL_REANALYSIS_PHY_001_025).

### Indicators for model skills

A set of diagnostics to rank the performance of the models were computed from the model outputs and the observational datasets for the period 1985–2005 for the Red Sea and the Gulf of Aden (Outer Region) separately (Fig 1, S1 and S2 Tables). Those diagnostics have been chosen to assess the skills of the models in reproducing different aspects of the heat content variability, which is what will determine the evolution of Red Sea temperatures under the projected global warming. The detailed formulation of each diagnostic is included in the Supplementary Information. All the diagnostics have been performed into the native model grid. However, for ensemble averaging or mapping all model fields have been bilinearly reinterpolated into a common grid. For a clearer visualization, we have chosen the KAUST model grid as the reference grid, as it is the finest one. In order to fill the coastline gaps in GCMs (i.e. due to their coarse resolution), we have extrapolated the values from the closest GCM grid point.

**Area and volume.** The spatial resolution of GCMs ranges from 0.2˚ to 2˚, so their representation of the Red Sea bathymetry may not be satisfactory. To quantify that we computed the area and volume of the Red Sea as represented by each model grid, and compare to those derived from the GEBCO database (General Bathymetric Chart of the Oceans), available at a high spatial resolution of 1/60˚ (https://www.gebco.net/data_and_products/gridded_bathymetry_data). Models showing a difference in the area or volume larger than +25% respect to the observed values, area (volume) for the Red Sea (i.e., ~4.57 10^5 km2 (2.18 ^5 km3)) and for the Outer Region (i.e., ~1.18 ^6 km2 (3.23 ^6 km2)), were discarded (see Section 3).

**Seasonal cycle of the averaged vertical profile.** A key element for a robust modelling of future Red Sea thermal regimes is the characterization of heat uptake at the sea surface and the heat transfer to deeper layers. In order to characterize the skills of the models on reproducing this, we computed the averaged vertical profile of seasonal temperature anomalies with respect to the annual average for the upper 400 meters and compared modelled and observed profiles. Seasonal comparisons of the averaged vertical profiles were based in terms of correlation coefficient and the root mean square error (henceforth RMSE), thus assessing the shape of the profile and the magnitude of the anomalies, respectively. The four values obtained (i.e., one for

each season) were then averaged to obtain the averaged seasonal correlation coefficient and RMSE.

**Seasonal cycle of the sea surface temperature.** The ability of the models to capture regional differences was assessed through their capacity to reproduce the spatial pattern of the seasonal anomalies. The focus is on sea surface temperature, where the largest regional differences are expected. In particular we compared modelled and observed seasonal anomalies in terms of spatial correlation and RMSE. The four values obtained (i.e., one for each season) were then averaged to obtain the averaged seasonal correlation coefficient and RMSE.

**Interannual variability.** In order to assess the model skills in reproducing interannual variability, we computed the annual mean of the basin-average temperature field at each vertical level. We then calculated the standard deviation (std) to obtain a vertical profile of interannual std, which quantifies the amplitude of interannual signals at different depths. The shape and the values of the profiles were compared with observed ones in terms of correlation and RMSE. We further assessed the accuracy of the spatial patterns of sea surface temperature interannual std, again comparing them with the observed patterns in terms of spatial correlation and RMSE.

**Trends.** An additional diagnostic that helps to identify potential problems with heat uptake in the models is the evaluation of trends in the recent decades. We cannot expect that GCM free runs reproduce realized decadal trends, so the goal of this diagnostic is to identify models that are drifting, so reproducing unrealistic trends (i.e. trends exceeding what could be expected by natural variability or climate change). In this case, we did not include the GLORYS and KAUST models in the assessment because they cover a different time window, and the trend computation is very sensitive to the length of the record used for trend estimates. The slope of the linear trend of the basin averaged SST ($\tau$) (i.e. the slope of a line fitted to the annual mean sea surface temperature), and its associated standard error ($\varepsilon$) were computed for each model and compared with those of TEMPERSEA. The difference in the trends between observations and model outputs ($\theta = |\tau_{mod} - \tau_{obs}|$) was compared with the square root of the quadratic sum of the standard errors of the fitting ($\varepsilon = \sqrt{\varepsilon_{mod}^2 + \varepsilon_{obs}^2}$). This has then be used to select suitable models (see section 3).

## Model skill assessment

### Area and volume

In the Red Sea, 32 of 43 models were in the range of $\pm > 25\%$ with respect to the actual area, and 24 of 43 with respect to the actual volume (Fig 2). In the Outer Region, 41 of 43 and 39 of 43 meet that condition, for area and volume, respectively. As a reference, the global model GLORYS reaches a 91% (90.2%) of the actual area (volume), while the regional high-resolution KAUST model represents a 100.7% (99.7%) of the actual area (volume). Three of the CMIP5 GCMs did not include any representation of the Red Sea, and 9 of those including the Red Sea represent it as a much smaller basin. Other 9 models show a reasonable surface area but were biased toward either a much shallower or deeper Red Sea, and were, accordingly, discarded. Conversely, in the Outer Region only 5 of those discarded models did not fulfil the requirements (see S1 Table).

### Seasonally averaged vertical temperature profile

In the Red Sea, the observed basin averaged temperature profile shows an anomaly with respect to the 3D basin average of ~2˚C at the surface during winter and spring (Fig 3, black line). In summer and autumn, the temperature anomaly exceeds 5˚C. Below 100 m the

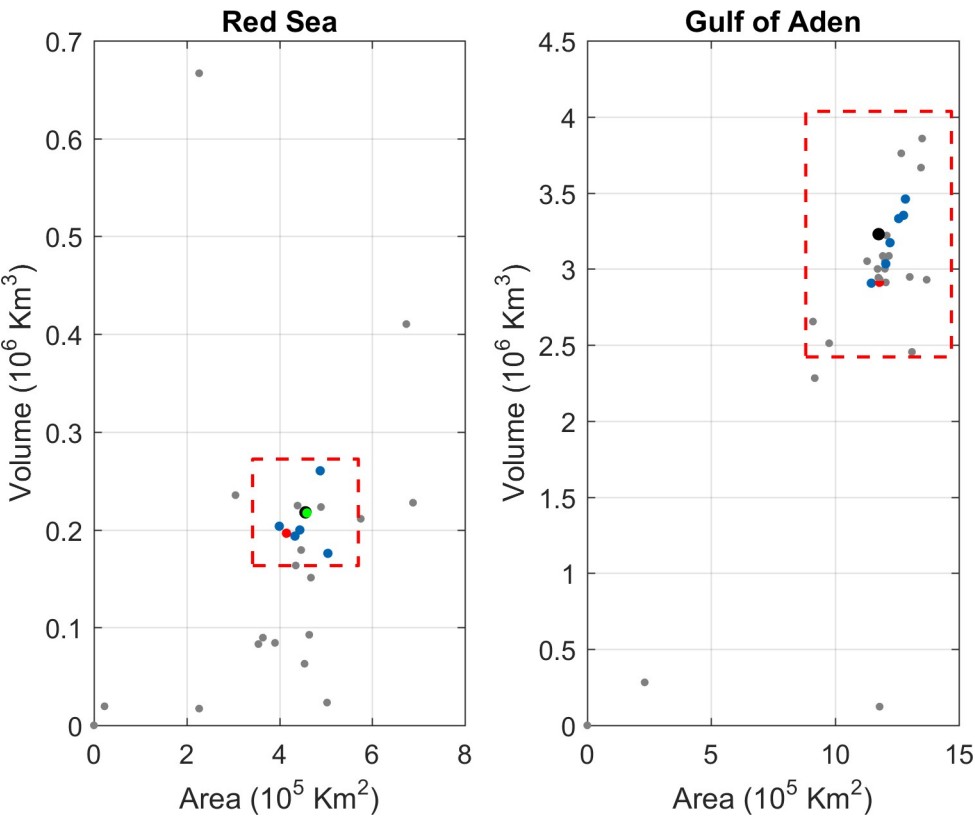

**Fig 2.** Scatter that represents area (x-axis) against volume (y-axis) for each GCM (grey dots) and the reference simulations (GLORYS in red and KAUST in green). The observed values are indicated by the black dots and the acceptble threshold (±25% with respect to the observed value) is represented by the dashed red line. The results are shown for the Red Sea basin (left panel) and the Gulf of Aden (right panel). Note that KAUST simulation does not cover completely the Gulf of Aden and is therefore not included in the right panel.

anomalies are constant throughout the year (~-2.5˚C). The shape of the profile also showed a seasonal evolution in the upper 100 m, being more homogeneous in winter and strongly stratified in summer and autumn. The reference simulations GLORYS and KAUST showed small RMSE (0.30˚C and 0.34˚C, respectively) and a good representation of the seasonal changes of the profile shape (averaged correlation of 0.99 for both hindcasts). Regarding the GCMs, two models deviated greatly from the rest: "ACCES 1–0" and "ACCES 1–3" (S1 Table). For the rest of models, the range of RMSE was 0.5˚C -3.5˚C, and the correlations fluctuated between 0.65 and 0.99 (Fig 3, S1 Table).

In the Outer Region the surface anomalies were larger in spring (up to +8˚C) and the smallest in winter (~+6.2˚C), when the mixed layer reached up to 80 m compared to the stronger stratification found in summer (S1 Fig). GLORYS showed good skills in reproducing the shape and the seasonal evolution of the anomaly profiles, with an averaged RMSE of 0.32˚C and a correlation of 0.99. Most GCMs models had a temperature anomaly between 1 and 5˚C lower than observations, from surface up to 100 meters of depth. Almost all the models exhibited between 1 and 5˚C higher temperature values than observed between 100 and 300 meters depth. Hence, most GCMs were unable to properly reproduce the heat uptake in the upper layers in winter and spring. In autumn and summer, the GCMs performed better but most models showed a reduced vertical gradient of temperature (S1 Fig).

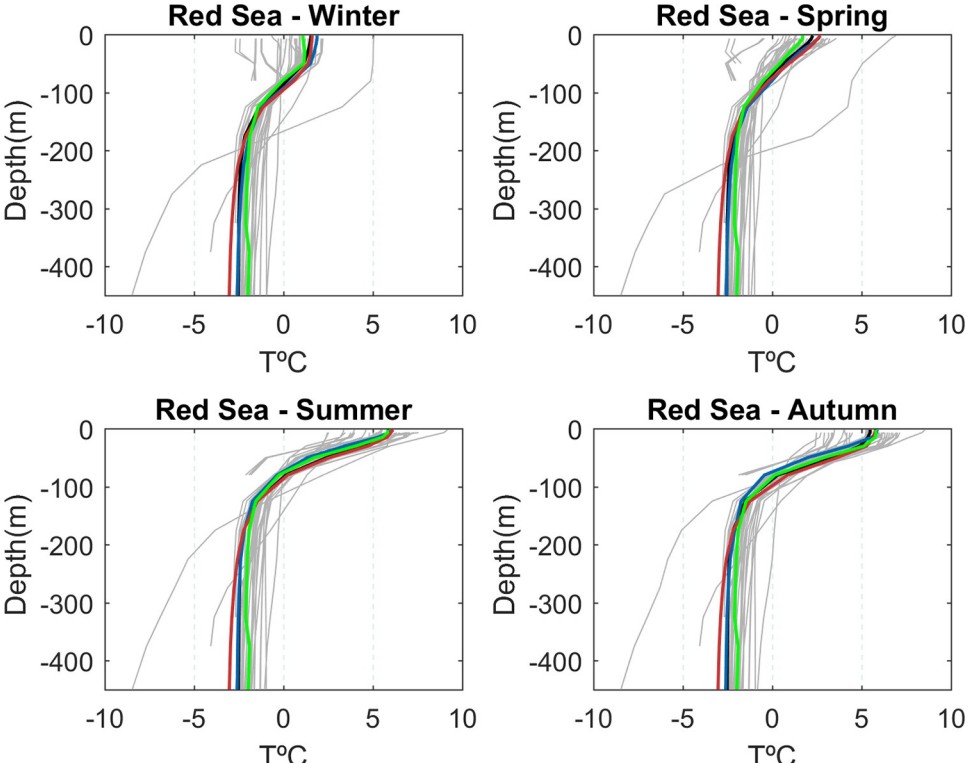

**Fig 3.** Seasonally averaged vertical profile of temperature anomalies in the Red Sea for the observations (black line), CMIP5 models (grey lines), GLORYS (blue line), KAUST (red line), and the average of the selected GCMs (see "Model Selection" section, green line).

## SST seasonality

The seasonal anomaly patterns are shown in Fig 4. In winter, the largest negative anomalies (up to -4˚C) in the Red Sea basin, relative to the basin-average annual mean, were found in the northern Red Sea, while the smallest anomalies (around -1˚C) were located around 18˚N (Fig 4). In summer the seasonal pattern is rather similar, with the lowest anomalies in the northern Red Sea (approximately 0˚C) and the highest in the southern part (around +4˚C). The reference runs show a very similar behaviour, with an averaged RMSE of 0.23˚C and 0.29˚C for GLORYS and KAUST, respectively and an averaged spatial correlation of 0.98 and 0.99 respectively. The GCMs show RMSE values ranging from 0.5˚C to 2.8˚C (see S1 Table), and spatial correlations ranging from -0.60 to 0.95. No significant differences in the model skills have been found for different seasons.

For illustrative purposes we focus on the results of two models (Fig 4). The MPI-ESM-LR model showed good skills in the Red Sea (RMSE 0.70˚C and correlation 0.88): the seasonal evolution is very well captured, the local winter maximum is in the right place around 18˚N, and the range of values also match well the observations in both seasons. On the other hand, GFDL-ESM2G showed one of the poorest skills across all models tested for this diagnostic in the Red Sea (RMSE 1.10˚C and correlation 0.69). This is mainly because of the spatial structure of the anomalies which show the local maximum was placed northwards. Also, the winter values were about 1˚C cooler than observed, while the summer values were about 1˚C warmer. However, the seasonal transition was well captured even in this case (see S1 Table).

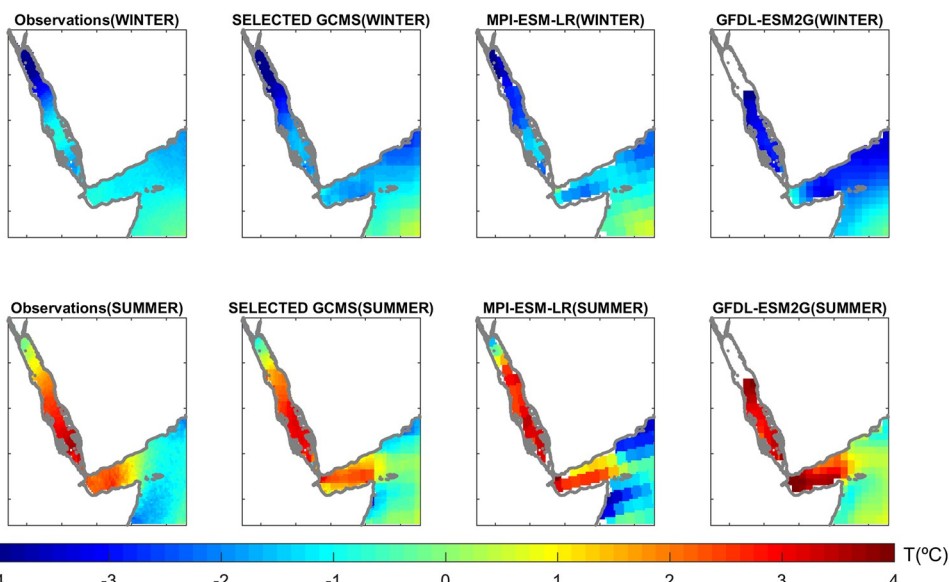

**Fig 4.** Sea surface temperature anomalies with respect to the annual average (in ˚C) in winter (Top) and summer (Bottom), for the observations (first column), average of the selected GCMs (see "Model Selection" section; second column), an example of GCM with good skills (MPI-ESM-LR; third column) and an example GCM with low skills (GFDL-ESM2G; last column).

## Amplitude of interannual variability at different depths

The observed vertical profile of the interannual std in the Red Sea ranges from 0.25˚C at 50 m to 0.05˚C below 300 m (S2 Fig, left panel). Interannual std was larger in the Gulf of Aden (0.30˚C at 75 m to 0.10˚C below 300 m), probably due to lateral advection of oceanic waters [17]. In the Red Sea, KAUST shows similar values, even if the shape of the profile is somehow different, with an RMSE and correlation of 0.045˚C and 0.89, respectively. Conversely, the other reference model, GLORYS, shows a very different behaviour, with a maximum reaching 0.6˚C at 75 m and values close to 0.2˚C at 300 m. This leads to a RMSE of 0.2˚C and a correlation of 0.69 in the Red Sea. In the Gulf of Aden, GLORYS also shows a wrong profile with a maximum of 0.55˚C at 125 m and minimum of 0.15˚C below 300 m, which is larger than the observed values (S2 Fig, right panel). Most GCM outputs represented the interannual variability reasonably well, both in the Red Sea and in the Gulf of Aden. The shape of the profile followed the observed one and maximum std values were below 0.3˚C in most cases, with almost all GCMs showing better skills than the reference GLORYS simulation (see S1 Table).

## Interannual variability of SST

The observed interannual variability of SST is higher than 0.4˚C in the north of the Red Sea while reaches minimum values of 0.2˚C in the southern part (Fig 5). The reference simulations, KAUST and GLORYS also show a northward gradient, although smoother and with values ranging from 0.25˚C to 0.35˚C. Their RMSE is 0.05˚C and their spatial correlation is 0.59 and 0.53, respectively. In the Red Sea, the GCMs show a wide spread of skills, with RMSE values ranging from 0.05˚C to 0.25˚C and spatial correlations ranging from -0.6 to 0.6. In the Gulf of Aden, the RMSE ranges from 0.08˚C and 0.22˚C and between -0.5 and 0.6 for the spatial correlation (see S1 Table). An example of a GCM showing good skills (IPSL-CM5a-LR) is presented in Fig 5 (RMSE of 0.07˚C and correlation 0.32). Its behaviour is close to the GLORYS pattern. It represents the northward gradient correctly, although smoother than the

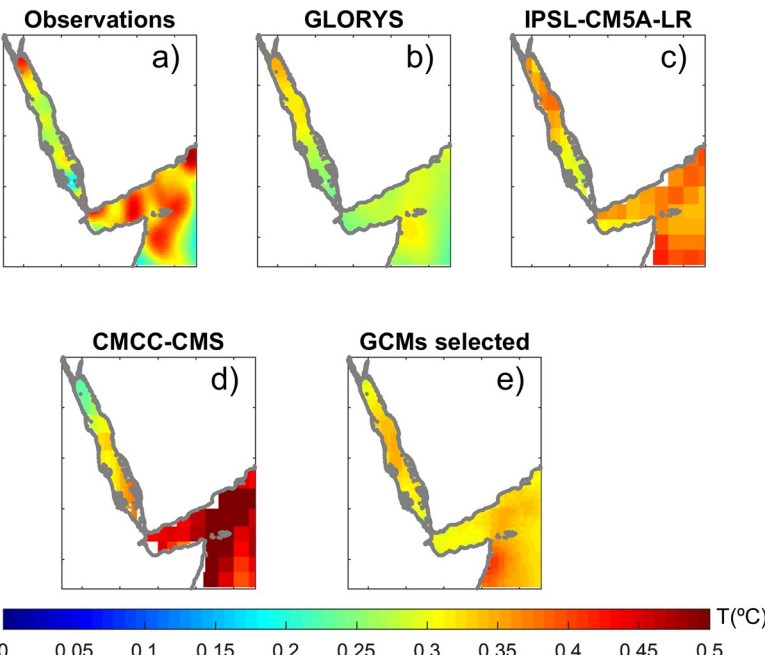

**Fig 5.** Interannual std of sea surface temperature from (a) observations, (b) the GLORYS reanalysis, (c) a GCM with good skills (IPSL-CM5a-LR;), (d) a GCM with low skills (CMCC-CMS;) and (e) the average of selected GCMs (see "Model selection" section).

observations. The range of values is between 0.25˚C to 0.4˚C, larger than GLORYS and closer to the observed range. Conversely, the CMCC-CMS model shows poor skills (RMSE of 0.09˚C and correlation -0.63).

## Trends

The interannual variations and the long term evolution of the basin averaged SST anomalies showed by the two reference simulations were in good agreement with the observations both in the Red Sea and the Gulf of Aden (Fig 6). The observed linear trends (±> the standard error) for the period (1985–2005) were 0.021 ±>0.007˚C year$^{-1}$ in the Red Sea and 0.002 ±>0.007˚C year$^{-1}$ in the Gulf of Aden. As KAUST and GLORYS do not cover the same period, trends were not computed for these datasets. The GCMs showed trends ranging from 0˚-C·year$^{-1}$ to 0.050˚C·year$^{-1}$ in both regions, with an average value of 0.021˚C (S1 and S2 Tables). GCMs showing trends outside the range defined by the observed trend plus ± the uncertainty (square root of the i.e. quadratic sum, l264), were flagged as suspicious.

## Model selection

The different diagnostics presented above had aimed at characterizing different aspects of the heat uptake, which is what determines the response of the Red Sea to the projected global warming. However, the quality of the GCMs was very heterogeneous, with some of them showing low skills in most of the diagnostics considered before. Therefore, prior to analyse their results for the climate projections, we have to select a subset of GCM models displaying reasonable skills in a selection of diagnostics. Unfortunately, there is no objective method that ensures the "right" choice of diagnostics, as some models perform well in some of them while performing poorly in others. Therefore, we defined 4 different performance criteria based on a

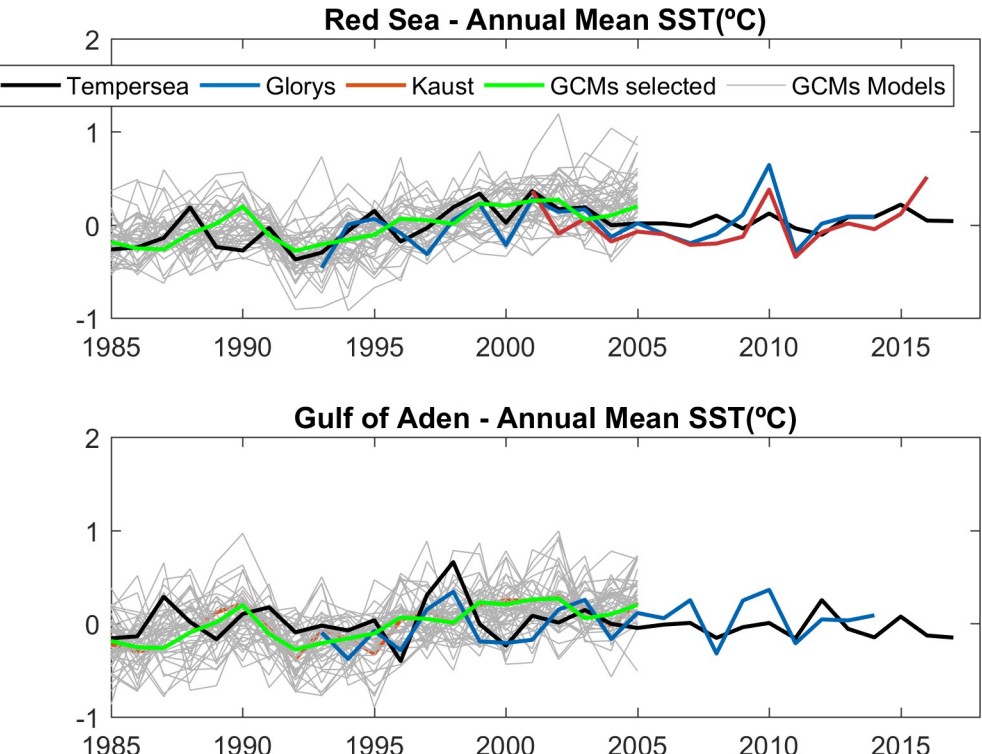

**Fig 6.** Annual Mean SST anomalies (°C) in the Red Sea (Top) and in the Gulf of Aden (bottom) for the observations (black line), KAUST (red line), GLORYS (blue line), GCMs (grey lines) and the average of the selected GCMs (see "Model Selection" section; green line). Anomalies are computed with respect to the 1985–2005 average.

selection of the diagnostics presented above (Table 1). In all cases, those models showing a bad representation of the basin topography in terms of surface and/or volume are rejected and not further discussed here. Then, for each criterion (i.e. subset of diagnostics), we discard those models that show a RMSE three times larger than the RMSE of the reference simulation in any of the selected diagnostics. Additionally, for some of the criteria those models showing trends outside the range defined by the observed trend plus ± minus the uncertainty were discarded.

Namely the different criteria are the following (Table 1):

Criterion 1: Models that fulfill all diagnostics.

Criterion 2: Models with a reasonable representation of the vertical structure and variability of the temperature field at seasonal and interannual time scales and showing a present climate trend compatible with observations are kept. I.e. in this criterion more weight is given to the vertical transfer of heat.

**Table 1. Definition of the diagnostics considered in each selection criterion: Area/Volume; Seasonal Average Vertical Profile Temperature (SAVPT); Seasonal Cycle of the Sea Surface Temperature (SCSST); Vertical profile of interannual std (STDVPT); Interannual std of the Sea Surface Temperature (STDSST) and long term Trends in the present climate.**

|      | Area/Vol | SAVPT | SCSST | STDVPT | STDSST | Trends |
|------|----------|-------|-------|--------|--------|--------|
| **Cr 1** | ✓ | ✓ | ✓ | ✓ | ✓ | ✓ |
| **Cr 2** | ✓ | ✓ | ✗ | ✓ | ✗ | ✓ |
| **Cr 3** | ✓ | ✗ | ✓ | ✗ | ✓ | ✓ |
| **Cr 4** | ✓ | ✓ | ✓ | ✓ | ✓ | ✗ |

**Table 2. List of GCMs that meet at least one of the criterion.** The last two columns indicate whether the model has been run under scenario RCP8.5 and RCP4.5. In bold, the models that have been selected for the analysis of temperature projections (Criterion 2 and results for the two scenarios).

| Model | Cr 1 | Cr 2 | Cr 3 | Cr 4 | RCP 8.5 | RCP 4.5 |
|---|---|---|---|---|---|---|
| **GISS-E2-R-CC** | ✓ | ✓ | ✓ | ✓ | ✓ | ✓ |
| **IPSL-CM5A-MR** | ✓ | ✓ | ✓ | ✓ | ✓ | ✓ |
| MIROC4H | ✓ | ✓ | ✓ | ✓ | ✗ | ✓ |
| **MPI-ESM-LR** | ✗ | ✓ | ✗ | ✗ | ✓ | ✓ |
| **MPI-ESM-MR** | ✗ | ✓ | ✗ | ✗ | ✓ | ✓ |
| **CMCC-CESM** | ✗ | ✓ | ✗ | ✗ | ✓ | ✓ |
| **CMCC-CM** | ✗ | ✓ | ✗ | ✗ | ✓ | ✓ |
| **CMCC-CMS** | ✗ | ✓ | ✗ | ✗ | ✓ | ✓ |
| MPI-ESM-P | ✗ | ✓ | ✗ | ✗ | ✗ | ✓ |
| **NORESM1-M** | ✗ | ✓ | ✗ | ✗ | ✓ | ✓ |
| CNRM-CM5-2 | ✗ | ✗ | ✗ | ✓ | ✗ | ✓ |
| IPSL-CM5A-LR | ✗ | ✗ | ✗ | ✓ | ✓ | ✓ |

Criterion 3: Models with a reasonable representation of the spatial structure of the surface variability at seasonal and interannual scales and showing a present climate trend compatible with observations are kept. I.e. in this criterion more weight is given to the spatial structure of the heat distribution.

Criterion 4: Models fulfilling all the criteria except the one on trends are kept. This will be a version of Criterion 1 without the constrain in the trends.

The models included following each criterion are listed in Table 2.

In this work, we are mainly interested in the evolution of the heat content of the basin (i.e. the averaged 3D temperature) and the vertical structure of the temperature field. So, we have followed Criterion 2 to do the GCM selection, as a compromise between model reliability and keeping enough models to do the projections. In other words, following Criterion 2 we sacrifice the quality in the description of the horizontal structure of the surface temperature field in order to keep enough models for robust ensemble averaging. There are 10 models fulfilling criterion 2, and 8 of them are available for the two RCP scenarios (Table 2, S3 Fig). As mentioned above, there is an unavoidable degree of subjectivity when selecting the criterion, so the implications of this choice on the final results are addressed in the Discussion section.

## Temperature projections

### Sea surface temperature

The ensemble of selected models projects Red Sea annual mean sea surface warming ranging between 2˚C and 4˚C by the end of the century (2080–2100) under the RCP8.5 scenario (Fig 7, Tables 3 and S3). The highest warming is projected by CMCC simulations (CMCC-CM, CMCC-CMS and CMCC-CESM) and IPSL-CM5A-MR, with projected values above 3.5˚C. In contrast, GISS-E2-RCC projects the smallest warming (around +2˚C) followed by NOR-ESM1-M (+2.5–3˚C). In most cases warming is predicted to be quite homogeneous across the Red Sea basin, except for the MPI simulations (MPI-ESM-LR and specially MPI-ESM-MR), which show a clear north to south gradient, with warming in the southern Red Sea being approximately 0.6˚C greater than in the northern part.

In order to obtain the absolute value of projected temperatures we added the projected anomalies to the observed mean annual temperatures averaged over the period 1985–2005. The ensemble average of SST under scenario RCP8.5 (Fig 8) shows a temperature gradient

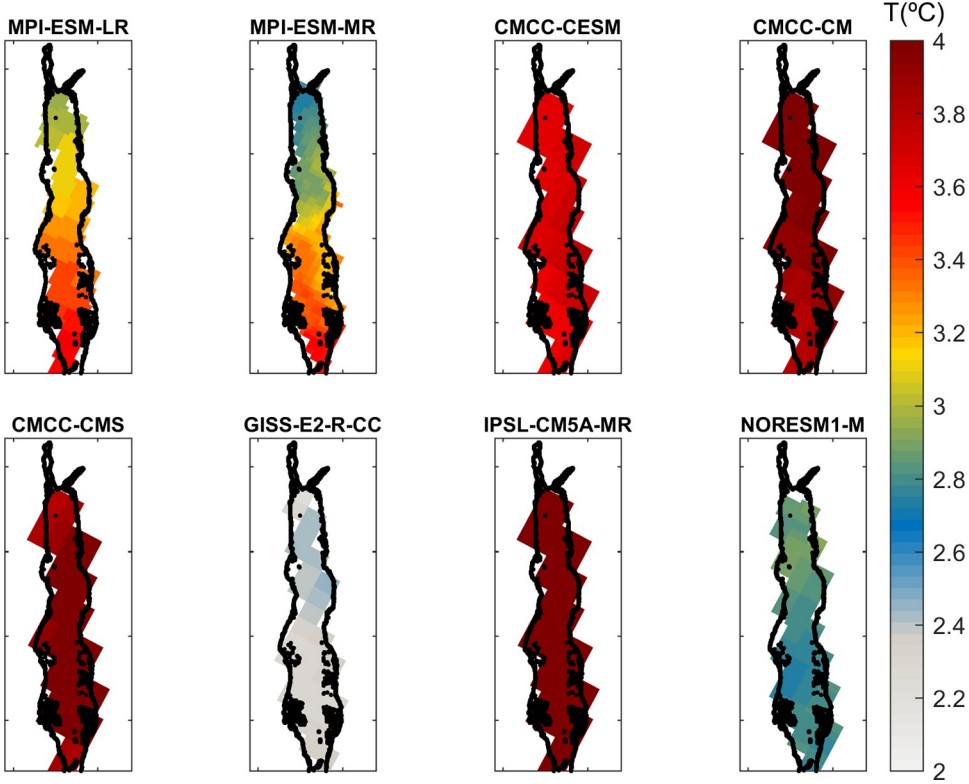

**Fig 7. SST increase at the end of the century (2080–2100) with respect to the present climate (1985–2005) under the RCP8.5 scenario for the selected GCMs (RCP 8.5).**

ranging from 29˚C in the northern part to 33˚C in the southern part. Under scenario RCP4.5 the spatial pattern is very similar, but the warming is lower, with values ranging from 26˚C in the north to 31˚C in the south. Additionally, we assess the uncertainty by looking at the ensemble spread. Due to the limited size of the ensemble (8 models), we evaluated the minimum and maximum values and the std from the ensemble (Fig 7, Table 3). Under scenario RCP8.5 the minimum projected values are 27˚C in the northern part while the maximum ones can be found in the southern part (34˚C). Under scenario RCP4.5, the minimum values would be 25˚C in the northern part and the maximum projected values are 31.5˚C in the southern part.

In order to analyse the rate of change throughout the century, linear trends have been fitted to the average surface temperature for two periods (Fig 9): 2006–2050 (period 1) and 2051–2100 (period 2). For the RCP 4.5 scenario, the trends are 0.23˚C/decade and 0.11˚C/decade, respectively. For RCP 8.5, the trends are 0.31˚C/decade and 0.47˚C/decade, respectively. Therefore, under scenario RCP4.5 there is a deceleration of the warming in the upper layers after 2050, while under scenario RCP8.5 is the opposite, with warming increasing after 2050.

**Table 3. Ensemble results for the basin averaged SST anomalies in the Red Sea (in ˚C) at the end of the century (2080–2100).** The ensemble mean and spread (STD) along with the ensemble minimum and maximum values are presented for scenario RCP4.5 and RCP8.5.

| | RCP4.5 (2080–2100) | | | | RCP8.5(2080–2100) | | | |
|---|---|---|---|---|---|---|---|---|
| | Ens. Mean | Ens. STD | Ens. Min | Ens. Max | Ens. Mean | Ens. STD | Ens. Min | Ens. Max |
| **Del_SST (˚C)** | 1.57 | 0.37 | 0.83 | 1.96 | 3.32 | 0.65 | 2.17 | 4.20 |
| **Del_T(0-225m) (˚C)** | 1.38 | 0.33 | 0.95 | 1.71 | 2.84 | 0.54 | 1.91 | 3.62 |

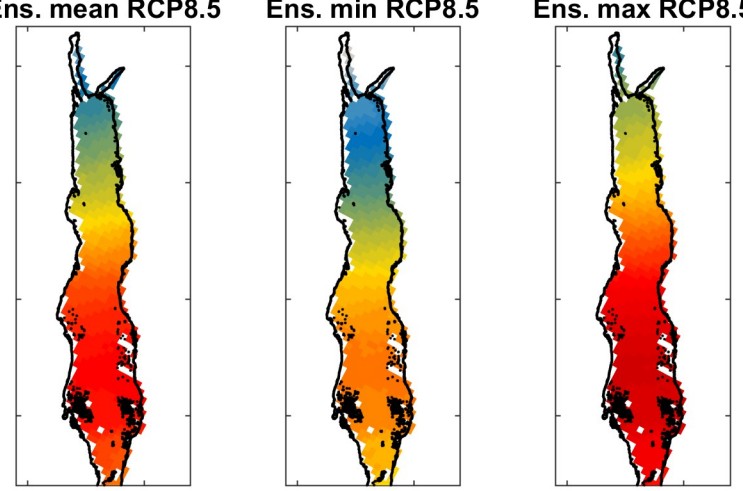

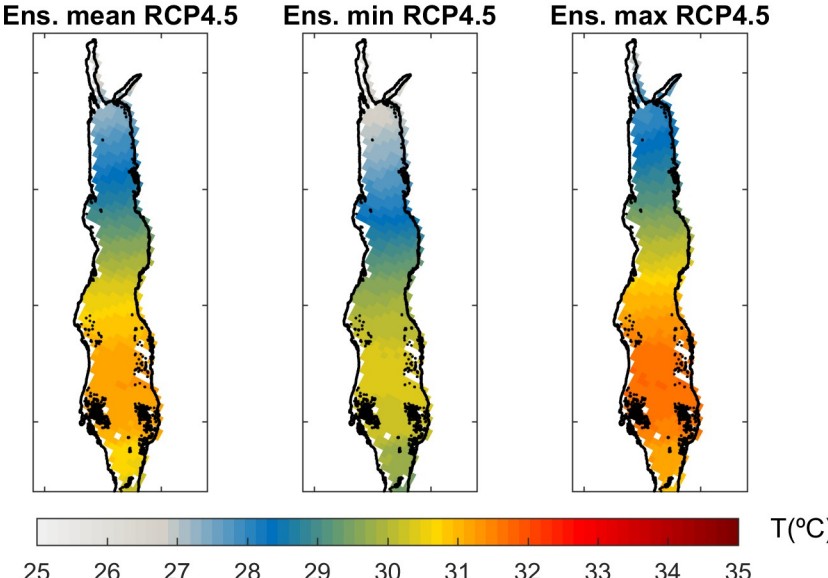

**Fig 8.** Projected SST (in ˚C) for the period 2080–2100 under RCP 8.5 scenario (top), and RCP 4.5 scenario (bottom). (Left column) Ensemble average, (middle column) ensemble minimum and (right column) ensemble maximum.

By the end of the century (2080–2100), the averaged surface temperature would increase with respect to present conditions 1.6˚C±0.4˚C˚C and 3.3˚C±0.7˚C (ensemble mean ±> spread) under scenarios RCP4.5 and RCP8.5, respectively (Fig 9).

## Temperature vertical profiles

All GCMs that have been evaluated project a rather uniform warming across the basin at different depths. Under scenario RCP8.5 they project more intense warming in the upper layers, with anomalies in the period 2080–2100 above 3˚C with respect to present values, while decreasing to 2.2˚C at 300 m (S4 Fig). This difference in the rate of warming between the two

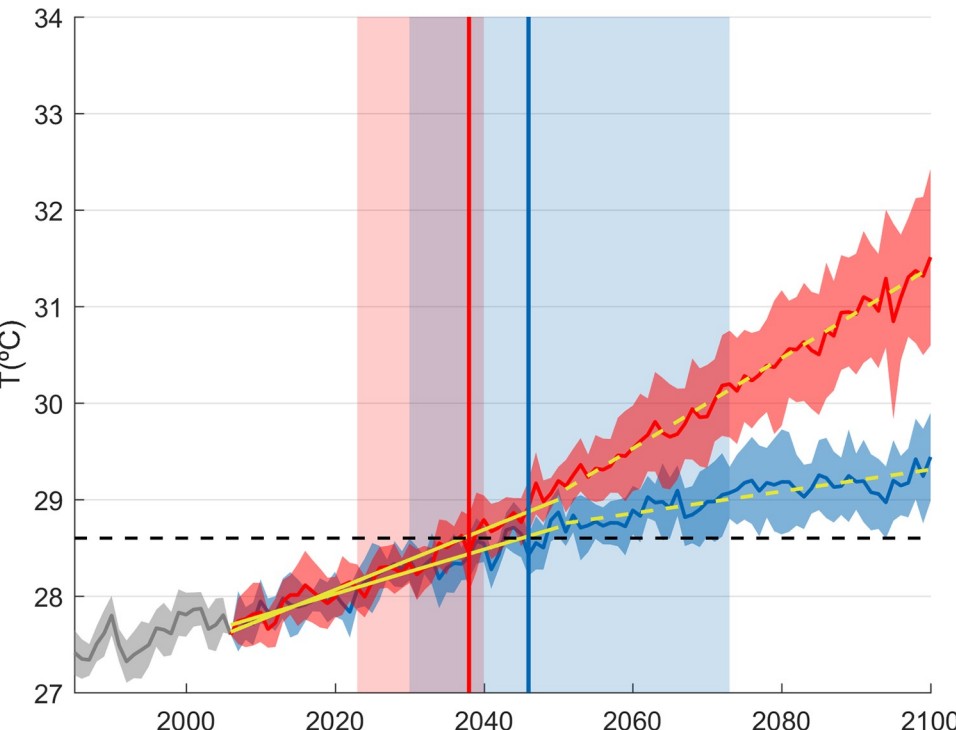

**Fig 9.** Historical (grey) and future projection of the Red Sea SST basin average (in ˚C) under scenario RCP 8.5 (red) and scenario RCP4.5 (blue). The solid lines indicate the ensemble average while the patch represents the range of values from the ensemble. Yellow lines indicates a linear trend fitted for both scenarios for the period 2006–2050 (solid lines) and 2051–2100 (dashed lines). The vertical boxes indicate the time period when the global temperature would be 1.5˚C above preindustrial levels in both scenarios.

depths will contribute to enhance the vertical stratification. The absolute value of the stratification is not available as the density fields were not available for all the models used in this study. However, the reported differential warming would contribute to increase the density difference between surface and 300m in 0.38 kg/m$^3$. Scenario RCP4.5 projects more vertically homogeneous warming, with projected anomalies of approximately 1.5˚C at all levels of the upper 300 m for the period (2080–2100). Under RCP4.5, thus, it is not expected that warming would induce changes in the stratification. The spread of model projections is almost constant through the water column, with an ensemble STD of 0.9˚C and 0.4˚C under RCP8.5 and RCP4.5, respectively. All models project the same behaviour in terms of the shape of the anomaly profiles under both scenarios, which provides an estimate of the robustness of the results, with enhanced stratification under scenario RCP8.5 and homogeneous warming under scenario RCP4.5 (S4 Fig).

## Impact of mean temperature rise on marine heat waves

The future evolution of marine heat waves depends on two factors. On the one hand it depends on changes in the mean temperature, and on the other hand on changes in the intra-seasonal variability. In terms of the probability density function (pdf), the first would imply a shift in the pdf location, while the second would imply a change in the shape of the pdf. The first factor is usually the dominant one, accounting for a large part of the projected changes in MHWs (e.g. in the Mediterranean the changes in the mean temperature account for more than 80% of the projected changes in the maximum temperature), [18,22]. Unfortunately, high frequency

outputs from all the selected GCMs were not available at the time this study was launched, so whether the mechanisms behind the extreme events will change could not be assessed. Nevertheless, recent studies have confirmed the results of [18] showing that the driver for MHW changes in most regions of the world is the change in the mean state, rather than changes in the intra-seasonal variability [7,38]. Therefore, here we focus on assessing how the changes in the mean state would modify the statistics of MHWs in the Red Sea and the Gulf of Aden.

To analyse how the increase of the mean temperature would affect the MHW characteristics in the Red Sea, we generate a high frequency projection of Red Sea SST for the period 2065–2100 by combining the observed present day intra-seasonal variability with the lower frequency components from the GCM outputs for the end of the century. Namely, we first compute daily maps of satellite SST anomalies for the present climate (1982–2017) removing the monthly average from the satellite data. Then, we add to these anomalies the monthly SST maps from each one of the selected GCMs (2065–2100). In other words, we keep the observed intra-seasonal variability while imposing the low frequency variability projected by the GCMs (see S5 and S6 Figs). Then, we compare the statistics of MHWs for that period (ensemble average) with the statistics of present day MHWs. It is worth noting that the intra-seasonal variability of the SST has a STD of 0.85˚C (see Skill Assessment Formulation in the SI), which is lower than the projected increase in the mean temperature. This, although not conclusive, reinforces the idea that changes in the MHWs will probably be dominated by the changes in the mean temperature.

In order to characterize MHWs at a particular location we follow the approach of [24], which is an adaptation for the marine environment of what is typically done to characterize atmospheric heat waves. Namely, a MHW is defined as a period of at least 5 consecutive days with temperatures above a certain percentile, computed from, at least, a 30-year period in the present climate. [24] proposes the use of the 90th percentile, while other studies use the 99th percentile [18,22,23]. Then, for the evaluation of future MHWs, the projected heat waves are defined following the same criterion, but using the threshold computed for the present climate. It must be noticed that the choice of any of those thresholds is somehow subjective and the MHWs characteristics could change depending on it. Therefore, we have tested the MWHs characteristics using different thresholds (90th,95th,98th and 99th), to assess the differences of the MHWs in the future with respect to present conditions. The results are qualitatively very similar (S7 Fig), so they will not be discussed here in detail. The results shown here are based on the definition of [28], which is an adaptation of the previous studies tuned to better represent the conditions that induce coral bleaching in the Red Sea. In particular, we consider a MHW as a period of at least 7 consecutive days with temperatures above the 95th percentile of the satellite SST computed for the period 1982–2017 at a particular location. This definition has been used to characterize the major events of coral bleaching in 1998, 2007, 2010, 2012 and 2015 [28].

The 95th percentile of present SST shows significant regional differences. In the Outer Region, the values are minimum (30˚C), while it increases to 31.5˚C in the western part, in the Gulf of Aden, less affected by the open sea dynamics and lateral mixing. The 95th percentile increases to 33˚C in the Southern Red Sea (18˚N) and decreases to 30˚C in the northern Red Sea (S8A Fig). With this threshold, the number of MHWs per year ranges between 0.6 and 1 events/year, with higher values in the northern part of the Red Sea and the Outer Region (S8B Fig). The mean intensity of the MHWs, defined as the average excess over the chosen threshold during the heat wave duration [24], tends to have larger values in the Outer Region and the Gulf of Aden, where intensities exceed 0.7˚C. In the rest of the domain, MHWs intensities are around 0.5˚C (S8C Fig).

Our results project a mean increase of the summer SST of approximately 3.3 + 0.6˚C (1.6 ±> 0.3˚C) at the end of the 21st century under the RCP8.5 (RCP4.5) scenario. This increase

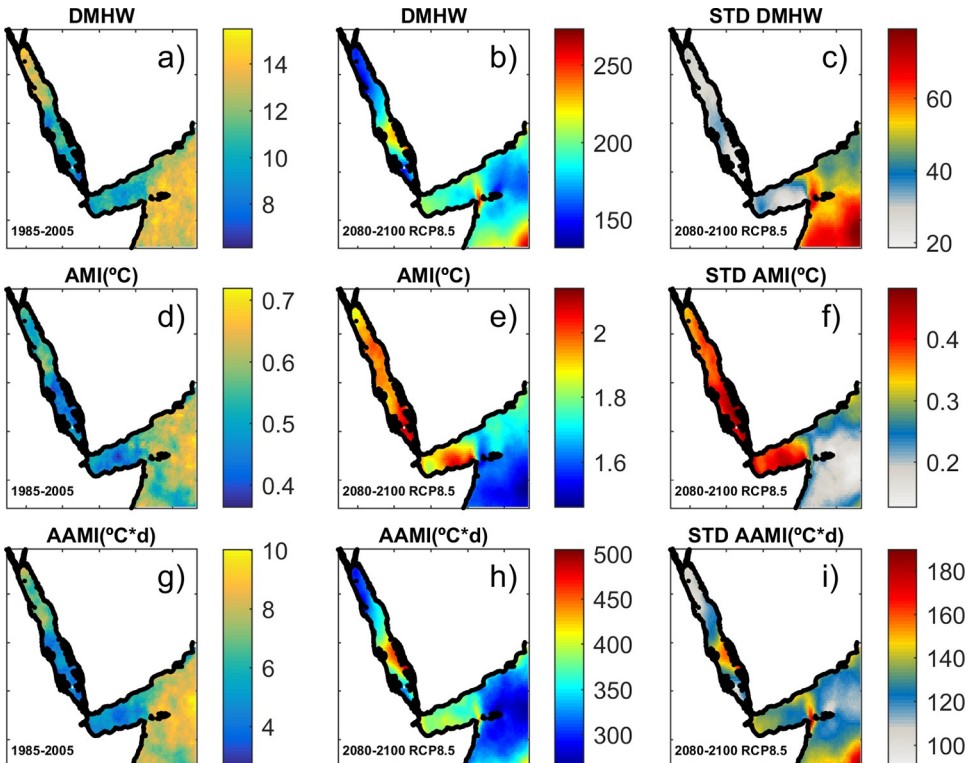

**Fig 10.** Statistics of MHWs for present conditions (left column), scenario RCP8.5 (ensemble average, middle column) and associated uncertainties (ensemble STD, right column). (a-c) Average length of MHWs in days per year. (d-f) Annual mean Intensity (˚C). (g-i) Annual Accumulative mean Intensity (˚C · day). Note the colorbars for the future scenarios are different than for the present conditions.

directly impacts on the MHW statistics. We analyse the future MHWs characteristics from the average of the 8 selected GCMs (Fig 10, mid column) and we use the ensemble standard deviation (STD) as a measure of the uncertainty (Fig 10, third column and S4 Table). The projections show that MHWs will happen every year, while the average duration of MHW, currently at 8–12 days per year (Fig 10A), will dramatically increase to around 120 ± 35 days per year in the south of the Red Sea and the Gulf of Aden, and around 50 ± 15 days per year in the rest of the region under the RCP4.5 scenario (S9B and S9C Fig). Under the RCP 8.5 scenario, MHW duration is expected to reach more than 200 ± 40 days per year in the South of the Red Sea and around 150 ± 20 days per year in the North, with values larger than 200 ± 60 days per year in the open ocean (Fig 10B and 10C).

The mean intensity of the MHWs will consequently increase. At present, the annual mean intensity (AMI) of MHWs ranges from 0.4 to 0.7˚C (Fig 10D), while it would increase to around 1–2 ± 0.20˚C under scenario RCP4.5 (S9E and S9F Fig) and to 2–3 ± 0.4˚C under scenario RCP 8.5 (Fig 10E and 10F). Under both scenarios, the spatial pattern of AMI is the same, with the largest values in the Red Sea and the lowest values in the outer region. Finally, we compute the annual accumulative mean Intensity (AAMI), defined as the accumulative excess of temperatures over the threshold. At present it is lower than 4˚C·day in the Red Sea and the Gulf of Aden, increases to 7˚C·day in the northern Red Sea and reaches a maximum of 10˚C·day in the open Indian Ocean (Fig 10G). This spatial distribution is mainly determined by the length of the MHWs (see Fig 10A). For the end of the century, the models project a large increase in the AAMI with a spatial pattern very similar under both scenarios, with the

largest values in the southern Red Sea and the lower values in the Outer Region. In particular AAMI would reach about 150 ± 35˚C·day under scenario RCP4.5, with maximum values in the southern Red Sea (180 ± 55˚C·day), decreasing to 50 ± 30˚C·day approximately in the Outer Region (S9H and S9I Fig). Under scenario RCP8.5 the projected AAMI is 300–400 ± 100˚C·day in the northern Red Sea and around 250 ± 120˚C·day in the Outer Region (Fig 10H and 10I).

## Discussion

GCMs are not designed to simulate regional processes, mainly because of their coarse resolution and the lack of specific tuning of model parameters, which is the main reason why GCMs are often discarded for regional analyses in semi enclosed seas [39]. Previous studies comparing regional and global models with observations suggest that regional simulations are required to properly simulate key processes in semi enclosed seas [16,40]. That would be the case for those analyses focused on the circulation and the 3D heat and salt distribution, which require a minimum spatial resolution to be properly solved. However, [18] showed that a relatively simple process like the air-sea interaction, can be reasonably reproduced by GCMs. Therefore, ocean variability which would only depend on the air-sea interaction could potentially be faithfully represented by GCMs. In this sense, [18] reported that projected changes of surface temperature in the Mediterranean for the 21st century were similar using an ensemble of GCMs or regional climate models. In conclusion, the suitability of GCMs for a particular study is case dependent. In order to further investigate the driving mechanism for the Red Sea heat content variability we compare observed heat content in the upper 100 m in the Red Sea from the TEMPERSEA product [17] with air temperatures at 1000 mbars obtained from the JRA55 reanalysis [16], (https://rda.ucar.edu/datasets/ds628.1/?hash=access). The comparison of the yearly values for the period 1958–2013 (Fig 11) shows that interannual variations in the Red Sea heat content are highly correlated to variations in the air temperature (r = 0.68). It has to be noted that both datasets are totally independent as one is constructed from ocean observations and the other using atmospheric data. This result reinforces the idea that heat content variations could be reproduced, at least as a first approximation, by models correctly reproducing air-sea interactions.

In order to assess if GCMs are able to reproduce the heat content variability in the Red Sea, here we have assessed the skills of 43 GCMs in terms of different diagnostics related to heat uptake. Our results are very heterogeneous with some of the models that did not even represent the Red Sea in their ocean mask, and others that showed very poor results in all the diagnostics. However, we have also found that some GCMs were able to reproduce the present variability with an accuracy comparable to the reference regional simulations (KAUST and GLORYS), especially inside the Red Sea. Those models are able to approximate reasonably well the seasonal and interannual variability of the temperature profile in the upper 300 m, suggesting that ocean heat uptake is properly modelled. Also, they are able to represent the spatial variability with the north to south gradients in the Red Sea and the distinctive behaviour of the Outer Region.

We found the skill of GCMs to be typically better inside the Red Sea than in the Outer zone. This is consistent with a previous study [17], that has shown that the heat content evolution in the Red Sea basin is mainly driven by the air-sea interactions, while the Outer Region is also affected by the advection of waters from the open Indian ocean. This suggests that the mechanisms underpinning the basin temperature evolution of the Red Sea are relatively simpler, and thereby easier to be captured by GCMs. In the Outer Region, GCMs should be able to properly model not only the air-sea interactions but also the temperature evolution and the circulation

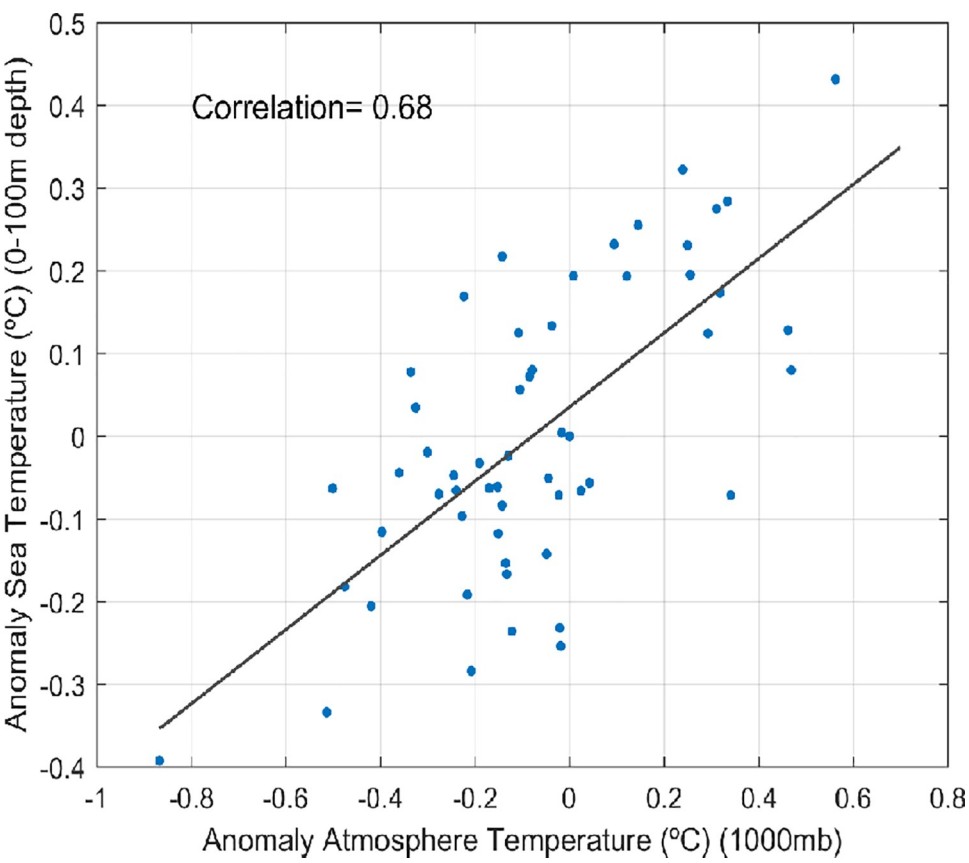

**Fig 11.** Air temperature at 1000 mbars (x-axis) and 0-100m sea temperature (y-axis) in the Red Sea. A high-pass filter has been applied to remove multidecadal variations.

patterns in the Indian Ocean. Thus, in our case, the isolation of the Red Sea from open ocean dynamics makes easier for the GCM to represent the heat content variability.

The temperature projections have been done based on the ensemble of 8 models that have met acceptable levels in selected diagnostics. Those diagnostics have been selected to reach a balance between the number of requirements and the number of models that are needed for the computations. However, as we have mentioned above, there is an unavoidable degree of subjectivity in any model selection procedure. Therefore, for completeness we have computed also the projected SST anomaly for different subensembles created following different criteria (Table 1). The results show that depending on the criterion chosen the projected SST increase under scenario RCP8.5 would range between 3.15˚C and 3.45˚C (S10 Fig). The former is obtained using criterion 1, the most restrictive, in which only 2 models complied (see Table 2). The latter was obtained using criterion 4, but in this case, there were no requirements on the GCM trends (Table 2). Namely, models with present climate trends larger than the observed range (including uncertainties) are kept and drive the ensemble towards larger projected anomalies. Therefore, the selected criterion (number 2), seems to provide a good compromise, between realistic representation of key processes and the number of models kept in the ensemble. Nevertheless, the final results would not have changed significantly if other criterion would have been selected (difference lower than 10%; see S10 Fig).

A note of caution is needed here. Although we think the selected GCMs provide a reasonably quality in reproducing Red Sea temperature variablity, the projections reported here have

to be considered as a first approximation. Higher resolution models would be able to reproduce more mechanisms and to provide more accurate projections. However, until an ensemble of high resolution ocean models is available the projections done by the selected CMIP5 GCMs can be considered as a benchmark result.

A recent study by [41], identified a multidecadal variability of Red Sea SST with a period of 70 years and an amplitude of approximately 0.3°C. Those authors suggested the Red Sea could enter into a cooling phase in the next years. However, that conclusion was derived from a rough extrapolation of the observations and can be reviewed in the light of the outcome of the more sophisticated approach presented here (S11 Fig). First, the observed multidecadal oscillation is overimposed to a long-term trend, so both components should be examined jointly. Depending on the long-term evolution during the next years, the absolute change will have a sign or another. Second, the concept of cooling requires the definition of a reference period. Overimposing the 70 years oscillation identified by [41] to the 20th century trend, the Red Sea SST would reach a relative minimum of 27.6°C in 2035–2040, 0.2°C lower than the value recorded during the 2000s but still higher than the mean value of the 20th century (27.5°C). However, this ignores global warming. If the 70 years oscillation is superimposed on the trend observed during the satellite period (1985–2019), the absolute temperatures would remain roughly constant till year 2035–2040 and then rise at an even higher rate due to the entering into the positive phase of the multidecadal oscillation. Finally, considering warming under "business as usual" RCP8.5 projected trends, the Red Sea SST will warm continuously even in the presence of a 70 years oscillation (S11 Fig).

The characterization of marine heatwaves is also of great importance for the Red Sea, where several episodes of coral bleaching have been reported in the past [29]. Due to the limited availability of high-frequency data for the selected GCMs, we have focused on the analysis of how the mean temperature change would affect the MHW characteristics. To do that we have projected the daily SST assuming that intraseasonal variability remains constant during the next decades (i.e. adding the projected monthly temperature anomaly to observed daily values). This has proven to be a reasonable assumption for the Mediterranean Sea, where only a 10–20% of the changes in MHW characteristics can be attributed to changes in the day-to-day variability [18,22]. Moreover, at global scale, [23] analysed the high frequency outputs from an ensemble of GCMs and reached the conclusion that the changes in the occurrence of MHWs are mainly driven by the global-scale shift in mean SSTs. This extent has also been confirmed by the observational study of [7,8].

Our study predicts a higher intensity of MHWs in the southern part of the Red Sea (Fig 10F and 10I), even if the warming pattern is similar throughout the region and the intraseasonal variability remains unchanged (see Supplementary Information). The reason for the sensitivity of the southern Red Sea to MHWs is found in the shape of the seasonal cycle, which is flatter in that region, resulting in an extended summer season. Therefore, once the mean summer temperature exceeds the local threshold, there would be many days per year exceeding that threshold. In the Northern Region or even more in the Outer Region, the summer is much shorter, so the increase in the mean value would have a more moderate impact on the MHW characteristics (Fig 12).

Finally, our results highlight the benefits of mitigation. The averaged warming in the sea surface would go from 3.3±0.6°C under a business as usual scenario to 1.6±0.4°C under a scenario of moderate mitigation (RCP4.5), with respect to period 1990–2010. Additionally, we can do a first assessment of what would be the situation in the Red Sea if the goals of the Paris agreement were reached and the global temperature increase with respect to preindustrial period remained below 1.5°C. That situation would happen under scenario RCP8.5 between years 2025–2040 (most likely 2038) and under scenario RCP4.5 between years 2040–2075

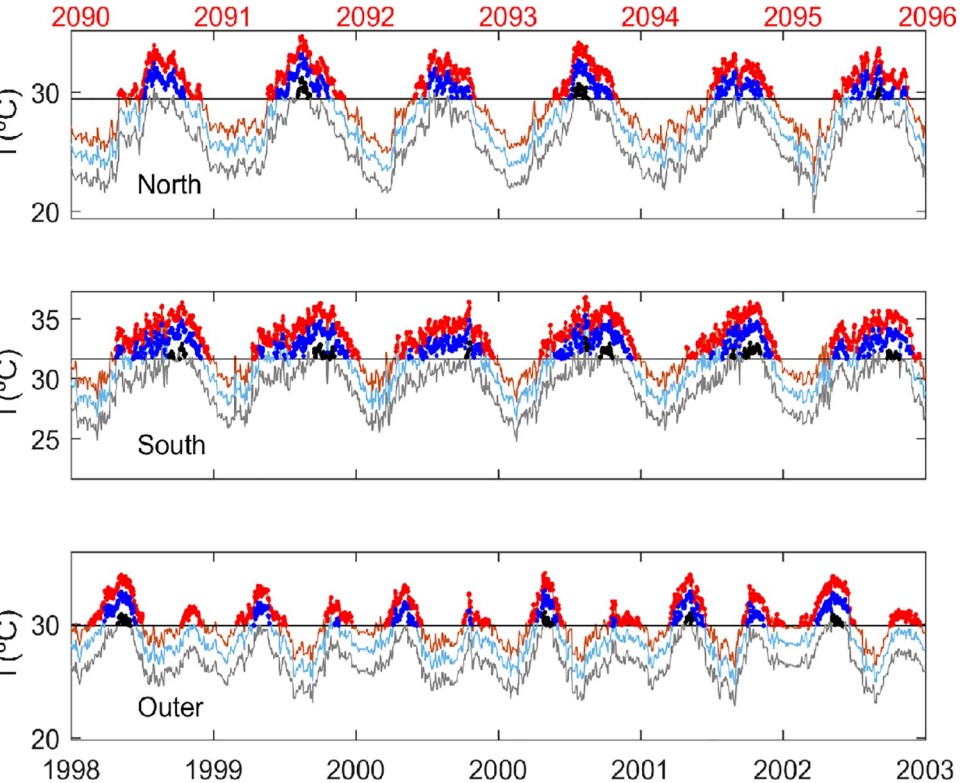

**Fig 12.** Daily SST averaged in the north (Top panel), south (middle panel) and Outer Region (bottom panel) for present conditions (grey), under scenario RCP 4.5 (light blue) and under scenario RCP8.5 (orange). The dots indicate the occurrence of MHWs in present conditions (black), under scenario RCP 4.5 (blue) and under scenario RCP 8.5 (red). For clarity only a limited period is plotted (1998–2003 for the present conditions, bottom axis and 2090–2096 for the future conditions, top axis).

(most likely 2045). In those periods and scenarios the SST increase in the Red Sea would be of only 0.7 ± 0.6˚C, which also matches the results obtained with 3 GCMs under scenario RCP2.6 by [13]. Therefore, it is clear that the expected impacts in the Red Sea in terms of mean warming but also in terms of MHWs would be much reduced under a scenario of strong mitigation.

## Conclusions

In this work we have assessed the skills of coarse resolution GCMs in reproducing the heat content variablity of the Red Sea. Namely we used different diagnostics on the seasonal, interannual and long-term variability of the temperature field. The results have shown very diverse results for the different GCMs, with almost half of them being unacceptable under the proposed criteria. However, we have also found a limited subensemble of 8 GCM models that show acceptable skills in most of the diagnostics, at a level comparable to two reference high resolution simulations. The selected subensemble was then used to project the temperature evolution in the Red Sea and the Gulf of Aden under a business-as-usual scenario (RCP8.5) and a moderate scenario (RCP4.5). The models project a larger warming under the former (3.3 ±0.6˚C at the surface for the period 2080–2100) than under the latter (1.6 ±0.4˚C at the surface for the period 2080–2100). Also, the models consistently showed that under scenario RCP8.5 the water column would become more stratified, while under scenario RCP4.5 the warming is expected to be quite homogenous in depth. This rise of the mean temperature would also affect the characteristics of marine heat waves. At present, the number of days with

MHW conditions ranges among 8 and 12 days per year in all the region. At the end of the century, under the RCP4.5 scenario, the number of days would increase to 125 days per year in the southern Red Sea and the Gulf of Aden, and around 100 days per year in the rest of the region. Under the RCP 8.5 scenario the situation would be even worse, reaching more than 200 days per year in the southern Red Sea and around 150 days in the northern part and the Gulf of Aden. Consequently, the annual mean intensity (0.6˚C) of the MHWs would increase in the Red Sea, being around 4 (3.6) times larger under the scenario RCP8.5 (RCP4.5). These projections based on GCMs have to be considered as a first approximation and a benchmark result until an ensemble of high resolution ocean models is available. Nevertheless, they clearly point towards the danger of a business-as-usual scenario of emissions and the great potential benefits that mitigation would bring to the Red Sea.

## Supporting information

**S1 Fig. Same as Fig 3 but for the outer region.** KAUST simulation does not cover this area and is therefore not included in the plots.
(TIF)

**S2 Fig.** Vertical profile of the interannual STD (in ˚C) in the Red Sea (left) and in the Gulf of Aden (right) for observations (black line), KAUST (red line), GLORYS (blue line), CMIP5 models (grey lines) and the average of selected GCMs (see "Model Selection" section, green line).
(TIF)

**S3 Fig. Maps of averaged SST anomaly for the period 1985–2005 (in ˚C) with respect to the basin average for each selected GCMs.**
(TIF)

**S4 Fig.** Left panel: Vertical profile of averaged temperatures in the Red Sea (in ˚C) for the observations (black), RCP4.5 (blue dashed lines, all models selected) and RCP8.5 (red dashed lines, all models selected) and the ensemble average is represented with a thick line. Right panel: Vertical profile of projected anomalies in the Red Sea for all the selected models under scenario RCP4.5 (blue) and RCP8.5(red). The ensemble average is represented with a thick line.
(TIF)

**S5 Fig.** (Top) Anomaly of the observed Sea Surface temperature (blue) and fitted seasonal cycle (red). (Bottom) Intra-seasonal variability of the observed sea surface temperature (grey).
(TIF)

**S6 Fig. Seasonal climatology of the selected CMIP5 models for Present (grey lines) and Future climate (red lines).** The values of each model are represented with thin lines, while the ensemble average is represented with thick lines.
(TIF)

**S7 Fig. Simulated changes in different MHWs descriptors presented as the ratio of projected values over the present climate values.** (a) annual mean duration, (b) annual mean intensity and (c) annual cumulative mean intensity (d). The colours represent different thresholds that have been used to define a MHW (90th, 95th, 98th and 99th percentiles).
(TIF)

**S8 Fig. Marine Heat Wave (MHW) characteristics in the present climate.** 95th -Percentile (left panel) in ˚C, number of MHWs per year (middle panel), and the annual mean intensity

(AMI) in ˚C (right panel).
(TIF)

**S9 Fig.** Statistics of MHWs for present conditions (left column), scenario RCP4.5 (ensemble average, middle column) and associated uncertainties (ensemble STD, right column). (a-c) Average length of MHWs in days per year. (d-f) Annual mean Intensity (˚C). (g-i) Annual Accumulative mean Intensity (˚C · day). Note the colorbars for the future scenarios are different than for the present conditions.
(TIF)

**S10 Fig. Projected SST anomaly for different subensembles according to different criteria (see Table 1).**
(TIF)

**S11 Fig. Simplified projections for the evolution of basin averaged SST.** Assuming the multidecadal oscillation identified by Krokos et al., (2019) continues along the 21st century, we test how the temperatures would change imposing different long term trends: The observed 20st century trend (in black), the trend observed during the satellite period (yellow) or the trend projected by models under scenario RCP8.5 (red).
(TIF)

**S1 Table. Set of diagnostics to rank the performance of the models from the model outputs and the observational datasets for the period 1985–2005, for the Red Sea and the Gulf of Aden (Outer Region) separately.**
(XLSX)

**S2 Table. Set of diagnostics to rank the performance of the models from the model outputs and the observational datasets for the period 1985–2005, for the Gulf of Aden (Outer Region).**
(XLSX)

**S3 Table. Change of the basin averaged sea surface temperature by the end of the century (2080–2100) respect to the present (1985–2005) of those GCM that fulfil at least one of the criterion (see Table 1).**
(XLSX)

**S4 Table. Statistics of MHWs for each selected GCM.** Average length of MHWs (DMHW, in days/year), Annual Mean Intensity (AMI, in ˚C/year) and Annual Acumulative Mean Intensity (AAMI, in ˚C x day/year). Results are presented for the present climate (1985–2015) and for projected future conditions under scenarios RCP4.5 and RCP8.5 for the period 2080–2100.
(XLSX)

## Acknowledgments

We thank marine biologists and physical oceanographers who have collected data and analysed the Red Sea mechanisms for the last decades. We specially acknowledge the support of the King Abdullah University of Science and Technology (KAUST), for providing data and knowledge about the temperature evolution and biology interactions of the Red Sea. We also thank Mediterranean Institute for Advanced Studies (IMEDEA, UIB/CSIC) and the Spanish Institute of Oceanography (IEO/CSIC) for providing the computational resources to address the goals of this work.

## Author Contributions

**Conceptualization:** Gabriel Jordà, Carlos M. Duarte.

**Data curation:** Miguel Agulles, Ibrahim Hoteit.

**Formal analysis:** Miguel Agulles.

**Funding acquisition:** Carlos M. Duarte.

**Investigation:** Miguel Agulles, Gabriel Jordà, Carlos M. Duarte.

**Methodology:** Miguel Agulles, Gabriel Jordà.

**Resources:** Carlos M. Duarte.

**Supervision:** Gabriel Jordà.

**Validation:** Miguel Agulles.

**Writing – original draft:** Miguel Agulles.

**Writing – review & editing:** Gabriel Jordà, Ibrahim Hoteit, Susana Agustí, Carlos M. Duarte.

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
