## [Decision Letter · Decision Letter 0]

31 May 2021

PONE-D-21-12932

Assessment of Red Sea temperatures in CMIP5 models for present and future climate

PLOS ONE

Dear Dr. AGULLES,

Thank you for submitting your manuscript to PLOS ONE. After careful consideration, we feel that it has merit but does not fully meet PLOS ONE’s publication criteria as it currently stands. Therefore, we invite you to submit a revised version of the manuscript that fully addresses all the points raised during the review process.

We look forward to receiving your revised manuscript.

Kind regards,

João Miguel Dias, Ph.D.

Academic Editor

PLOS ONE

Journal Requirements:

3. We note that Figures 1, 4, 5, 7, 8, 10, S4 and S7 in your submission contain map/satellite images which may be copyrighted. All PLOS content is published under the Creative Commons Attribution License (CC BY 4.0), which means that the manuscript, images, and Supporting Information files will be freely available online, and any third party is permitted to access, download, copy, distribute, and use these materials in any way, even commercially, with proper attribution. For these reasons, we cannot publish previously copyrighted maps or satellite images created using proprietary data, such as Google software (Google Maps, Street View, and Earth). For more information, see our copyright guidelines: http://journals.plos.org/plosone/s/licenses-and-copyright.

You may seek permission from the original copyright holder of Figures 1, 4, 5, 7, 8, 10, S4 and S7 to publish the content specifically under the CC BY 4.0 license. 

If you are unable to obtain permission from the original copyright holder to publish these figures under the CC BY 4.0 license or if the copyright holder’s requirements are incompatible with the CC BY 4.0 license, please either i) remove the figure or ii) supply a replacement figure that complies with the CC BY 4.0 license. Please check copyright information on all replacement figures and update the figure caption with source information. If applicable, please specify in the figure caption text when a figure is similar but not identical to the original image and is therefore for illustrative purposes only.

4. Please include captions for ALL your Supporting Information files at the end of your manuscript, and update any in-text citations to match accordingly. Please see our Supporting Information guidelines for more information: http://journals.plos.org/plosone/s/supporting-information.

Reviewers' comments:

Reviewer's Responses to Questions

**Comments to the Author**

1. Is the manuscript technically sound, and do the data support the conclusions?

Reviewer #1: Yes

Reviewer #2: Yes

2. Has the statistical analysis been performed appropriately and rigorously? 

Reviewer #1: No

Reviewer #2: Yes

3. Have the authors made all data underlying the findings in their manuscript fully available?

Reviewer #1: Yes

Reviewer #2: Yes

4. Is the manuscript presented in an intelligible fashion and written in standard English?

Reviewer #1: Yes

Reviewer #2: Yes

5. Review Comments to the Author

Reviewer #1: The work aims to assess the future projection of the Red Sea temperature in the global warming scenarios of the CMIP5 project. This topic is very relevant and has been poorly studied before. In this sense, I recommend publishing this work in order to increase our knowledge of the GCMs skill to represent the mean state and variability of the temperature field in the Red Sea, as well as the possible consequences that global warming can have in this region. However, there are major comments that should be addressed before publication.

The statistical analysis performed by the authors when assessing changes in Marine Heat Waves must be improved. There is no analysis of the uncertainty, robustness and agreement in models projections for this analysis and the authors only describe the results of the model ensemble average.

I feel that the document is a little disorganized. Most of the figures do not have proper titles or the units of the color palettes, what makes difficult to read them.

Specific comments are available at the joint document.

Reviewer #2: This paper analyses GCM performance in simulating Red Sea temperature, based on which the future projections are made from a subset of models with good performance. The authors have carried out a great deal of detailed work and their results are very reliable. The writing is also very clear and I could barely find any error. I believe the information provided by this paper will be valuable in terms of understanding climate change in this region.

6. PLOS authors have the option to publish the peer review history of their article (what does this mean?). If published, this will include your full peer review and any attached files.

Reviewer #1: **Yes: **Juan Pablo Sierra

Reviewer #2: No

---

## [Author Response · Author response to Decision Letter 0]

24 Jun 2021

We thank the editor and referee’s comments and the effort they made in reviewing our work. In the new version of the manuscript we have implemented all the points raised in the review. Thanks to those suggestions, we believe the new version of the manuscript has been significantly improved. 

In particular, we have updated the figures that needed to be improved, according to the reviewer’s comments. Figures corresponding to the results of each diagnostic have been updated adding the results of the ensemble of the selected GCMs . To clarify purpose, the titles and captions of those figures have been also updated. 

In order to address the lack of uncertainty in the MHWs characteristics provided for the future scenarios, we have calculated the projected changes of the MHWs characteristics for each GCMs individually. Then we use the ensemble standard deviation as a measure of the uncertainty for each diagnostic. Besides, we are aware that the supplementary figures were a bit disorganized, so we have re-ordered them to be consistent with the text. 

Finally, the rest of the detailed comments have been carefully addressed to fulfill the reviewer’s requirements.

---

## [Decision Letter · Decision Letter 1]

19 Jul 2021

Assessment of Red Sea temperatures in CMIP5 models for present and future climate

PONE-D-21-12932R1

Dear Dr. AGULLES,

We’re pleased to inform you that your manuscript has been judged scientifically suitable for publication and will be formally accepted for publication once it meets all outstanding technical requirements.

Kind regards,

João Miguel Dias, Ph.D.

Academic Editor

PLOS ONE

Additional Editor Comments (optional):

Reviewers' comments:

Reviewer's Responses to Questions

**Comments to the Author**

1. If the authors have adequately addressed your comments raised in a previous round of review and you feel that this manuscript is now acceptable for publication, you may indicate that here to bypass the “Comments to the Author” section, enter your conflict of interest statement in the “Confidential to Editor” section, and submit your "Accept" recommendation.

Reviewer #1: All comments have been addressed

2. Is the manuscript technically sound, and do the data support the conclusions?

Reviewer #1: Yes

3. Has the statistical analysis been performed appropriately and rigorously? 

Reviewer #1: Yes

4. Have the authors made all data underlying the findings in their manuscript fully available?

Reviewer #1: Yes

5. Is the manuscript presented in an intelligible fashion and written in standard English?

Reviewer #1: Yes

6. Review Comments to the Author

Reviewer #1: The authors have addressed all major comments and this interesting work, in my opinion, is ready to be published.

7. PLOS authors have the option to publish the peer review history of their article (what does this mean?). If published, this will include your full peer review and any attached files.

Reviewer #1: No

---

## [Editor Report · Acceptance letter]

22 Jul 2021

PONE-D-21-12932R1 

Assessment of Red Sea temperatures in CMIP5 models for present and future climate 

Dear Dr. Agulles:

I'm pleased to inform you that your manuscript has been deemed suitable for publication in PLOS ONE. Congratulations! Your manuscript is now with our production department. 

Kind regards, 

on behalf of

Prof. João Miguel Dias 

Academic Editor

PLOS ONE